# Systematic Review of Beta-Lactam vs. Beta-Lactam plus Aminoglycoside Combination Therapy in Neutropenic Cancer Patients

**DOI:** 10.3390/cancers16101934

**Published:** 2024-05-19

**Authors:** Kazuhiro Ishikawa, Tomoaki Nakamura, Fujimi Kawai, Erika Ota, Nobuyoshi Mori

**Affiliations:** 1Department of Infectious Diseases, St. Luke’s International Hospital, Tokyo 104-8560, Japan; morinob@luke.ac.jp; 2Department of Pulmonary Medicine, Thoracic Center, St. Luke’s International Hospital, Tokyo 104-8560, Japan; tonaka@luke.ac.jp; 3Library, Department of Academic Resources, St. Luke’s International University, Tokyo 104-0044, Japan; 4Global Health Nursing, Graduate School of Nursing Sciences, St. Luke’s International University, Tokyo 104-0044, Japan; otae@luke.ac.jp; 5Tokyo Foundation for Policy Research, Tokyo 106-0032, Japan

**Keywords:** beta-lactam plus aminoglycoside, combination therapy, febrile neutropenia, cancer patient

## Abstract

**Simple Summary:**

Febrile neutropenia is associated with high mortality, and the administration of anti-pseudomonal agents is required. Traditionally, combination therapy with aminoglycosides was weakly recommended in the guidelines, but recently, with the emergence of resistant bacteria, there has been increasing hope for combination antimicrobial therapy. The purpose of this systematic review, based on RCTs published up to 2023, is to examine how much more effective combination therapy with beta-lactams and aminoglycosides is compared to monotherapy with beta-lactams.

**Abstract:**

We performed a systematic review of studies that compared beta-lactams vs. beta-lactams plus aminoglycosides for the treatment of febrile neutropenia in cancer patients. Method: We searched CENTRAL, MEDLINE, and Embase for studies published up to October 2023, and randomized controlled trials (RCTs) that compared anti-Pseudomonas aeruginosa beta-lactam monotherapy with any combination of an anti-Pseudomonas aeruginosa beta-lactam and an aminoglycoside were included. Result: The all-cause mortality rate of combination therapy showed no significant differences compared with that of monotherapy (RR 0.99, 95% CI 0.84 to 1.16, high certainty of evidence). Infection-related mortality rates showed that combination therapy had a small positive impact compared with the intervention with monotherapy (RR 0.83, 95% CI 0.66 to 1.05, high certainty of evidence). Regarding treatment failure, combination therapy showed no significant differences compared with monotherapy (RR 0.99, 95% CI 0.94 to 1.03, moderate certainty of evidence). In the sensitivity analysis, the treatment failure data published between 2010 and 2019 showed better outcomes in the same beta-lactam group (RR 1.10 [95% CI, 1.01–1.19]). Renal failure was more frequent with combination therapy of any daily dosing regimen (RR 0.46, 95% CI 0.36 to 0.60, high certainty of evidence). Conclusions: We found combining aminoglycosides with a narrow-spectrum beta-lactam did not spare the use of broad-spectrum antibiotics. Few studies included antibiotic-resistant bacteria and a detailed investigation of aminoglycoside serum levels, and studies that combined the same beta-lactams showed only a minimal impact with the combination therapy. In the future, studies that include the profile of antibiotic-resistant bacteria and the monitoring of serum aminoglycoside levels will be required.

## 1. Introduction

Between 10 to 50% of patients with solid tumors and over 80% of individuals with hematologic malignancies develop febrile conditions as a result of chemotherapy-induced neutropenia [1]. This condition is especially prevalent among patients with blood-related cancers, who face a significantly elevated risk of febrile neutropenia, leading to potentially serious bloodstream infections primarily caused by gram-negative bacilli. The 2011 Infectious Diseases Society of America (IDSA) guidelines proposed monotherapy with an antipseudomonal beta-lactam for febrile neutropenia treatment [1]. Furthermore, the most recent European Conference on Infections in Leukaemia (ECIL) guidelines recommend escalation or de-escalation strategies based on the clinical presentation and individual risk of infections due to resistant strains of gram-negative bacilli [2]. Consequently, the use of broad-spectrum beta-lactam antimicrobial agents, including piperacillin/tazobactam (PIPC/TAZ), cefepime (CFPM), and carbapenems, is supported. These medications have been shown to be effective in addressing a broad spectrum of bacterial infections, which are of particular concern for patients with hematological malignancies during periods of reduced immunity due to chemotherapy [1,2,3].

A prospective study including patients with bloodstream infection due to gram-negative bacilli found reduced mortality in the subgroup of neutropenic patients who received empirical combination therapy with a beta-lactam plus an aminoglycoside [4]. However, a meta-analysis did not show a greater benefit with combination therapy than with beta-lactam alone in cancer patients with neutropenia [5]. Monotherapy for febrile neutropenia caused by high-risk multi-drug resistant gram-negative strains increases the frequency of carbapenem use and the risk of carbapenem resistance in Enterobacteriaceae (carbapenem-resistant Enterobacteriaceae, CRE). CRE infection has been reported in patients with hematologic tumors and is associated with a high mortality rate [6]. The administration of a non-carbapenem beta-lactam plus an aminoglycoside could be a potential treatment option for infections with high-risk multi-drug resistant-strains of gram-negative bacilli. A recent Spanish study reported that in febrile neutropenic patients with bacteremia, the beta-lactam plus aminoglycoside combination was associated with lower mortality and no difference in renal dysfunction compared to beta-lactam monotherapy [7]. A previous Cochrane review published in 2013 by M. Paul, et al. did not show the superiority of beta-lactam plus aminoglycoside combination therapy [5]; however, many of the randomized control trials (RCTs) included no anti-*Pseudomonas aeruginosa* antimicrobials. Therefore, we aimed to conduct a systematic review of beta-lactam plus aminoglycoside combination therapy versus beta-lactam therapy in cancer patients with neutropenia.

## 2. Materials and Methods

### 2.1. Objectives

The objective of this study is to evaluate and compare the safety outcomes associated with beta-lactam plus aminoglycoside combination therapy versus beta-lactam antimicrobial monotherapy in cancer patients experiencing febrile neutropenia. Specifically, the safety assessments will include all-cause mortality, infection-related mortality (as defined by individual RCTs), instances of treatment failure, and the incidence of renal failure (classified into the daily dose). This comparison aims to identify the most effective and safe treatment protocols for managing febrile neutropenia in cancer patients, thereby guiding clinical decisions and improving patient care outcomes in this vulnerable population.

### 2.2. Sources and Searches

An investigator developed the search strategies. Three databases, namely CENTRAL, MEDLINE, and Embase, were searched from 1 January 2012 until 30 October 2023. All search strategies are detailed in Appendix A, and the search terms used were similar to those used in the previous systematic review by M. Paul, et al. [5]. This systematic review was conducted according to the guidelines of the Preferred Reporting Items for Systematic reviews and Meta-Analyses (PRISMA) [8]. The review protocol was recorded on 8 December 2022, on PROSPERO, with the CRD number 42022379480.

### 2.3. Selection of Studies

We included both individual and cluster RCTs in any language where beta-lactam plus aminoglycoside combination therapy was administered to cancer patients with febrile neutropenia. The controls were administered beta-lactam antimicrobial monotherapy. Two investigators (K.I. and T.N.) independently assessed the full texts of the articles. We employed a two-stage screening strategy: The first stage involved removing duplicate articles from the three databases using EndNote software version 20.6 (accessed on 6 May 2024)(Clarivate Analytics), as well as excluding non-clinical trials by reading the title and abstract of each article. The second stage entailed a full-text review to extract randomized controlled trials (RCTs) that investigated our specified intervention and control. Discrepancies were discussed with a third investigator. Regarding the selection of the studies until the end of 2013, we referred to the results of the Cochrane review by M. Paul, et al. [5].

### 2.4. Outcomes

The primary outcomes in this systematic review were all-cause mortality, infection-related mortality (defined by each RCT), treatment failure, and adverse events of renal failure. The definition of treatment failure differed across studies. Details of the extracted studies are provided in Appendix A. Adults (older than 18 years) and children (younger than 18 years) with febrile neutropenia caused by cancer chemotherapy and treated with any antimicrobial regimen were included in this study. We defined fever as a single oral temperature higher than 38.3 °C or a temperature higher than 38.0 °C sustained for more than one hour, according to the guidelines [1,2]. Neutropenia was defined as an absolute neutrophil count of less than 500 cells/μL.

### 2.5. Data Extraction

Two investigators independently extracted the following data: year of publication, published country, sample size of each arm, type of cancer (solid tumor or hematological malignancy, including stem cell transplantation), type of beta-lactam antimicrobials and aminoglycoside, definition of infection-related mortality (although many studies did not provide a definition), treatment failure, and description of the resistant strain. Data were transferred to the data extraction sheet using the software Microsoft Excel version 16.84 (accessed on 6 May 2024) and Google spreadsheets (accessed on 6 May 2024). The following data were then checked by the reviewer: study information (e.g., publication country, study years, and single-center or multi-center study), participant baseline characteristics (type of population, inclusion and exclusion criteria, comorbidity, and type of cancer), information regarding the intervention (type of antimicrobials, aminoglycoside dosage, duration of administration of aminoglycoside, definition of renal failure, and whether serum aminoglycoside levels were measured in each RCT in Appendix A), information regarding the risk of bias (e.g., randomization method, allocation concealment, blinding, discontinuation of study, and incomplete outcome reporting), and information regarding outcomes (mortality, clinical failure, and renal failure). We preferentially extracted data using the intention-to-treat method, which included all individuals who were randomly assigned to the study outcome. For dichotomous outcomes, we recorded the number of participants manifesting the outcome in each group as well as the number of evaluated participants. For continuous outcomes, we documented values as well as the measures used to represent the data (including mean with standard deviation and median with interquartile range). We asked study authors for any missing data so that we could include findings from any studies published after 2012 in our study. Until 2012, we referred to the results of the Cochrane review by M. Paul, et al. [5]. Discrepancies were resolved via discussion with another investigator.

### 2.6. Risk of Bias Assessment

Two investigators independently assessed the risk of bias. Disagreements were resolved via discussion with a third investigator. The risk of bias was assessed according to the scales of the Cochrane risk-of-bias tool (RoB) [9]. Using RoB, we evaluated five domains of bias: random sequence generation, allocation concealment, blinding of participants and personnel, blinding of outcome assessment, and incomplete outcome data. We assessed the effect of allocation concealment on the results based on evidence of a strong association between poor allocation concealment and overestimation of the effect [10], as defined below:Low risk of bias (adequate allocation concealment)Unclear bias (uncertainty regarding allocation concealment)High risk of bias (inadequate allocation concealment)

The two review authors independently recorded the methods of allocation generation, blinding, incomplete outcome data, and selective reporting; the unit of randomization (patient or febrile episode); and the publication status; in addition to the adequacy of allocation concealment.

### 2.7. Statistical Analyses

We analyzed continuous and dichotomous data by calculating the risk ratio (RR) for each study with the uncertainty in each result presented as 95% confidence intervals (95% CIs). A fixed-effects model was used unless significant heterogeneity was observed (*p* < 0.1 or I^2^ > 50%), where the random-effects model was used. We analyzed the data using Review Manager 5.4 software, released by Cochrane. In our systematic review, we studied a mixed population, including children and adults, spanning various ages from earlier to more recent studies. Therefore, we conducted sensitivity analyses based on age and year of publication.

### 2.8. Certainty of Evidence

We used the Grades of Recommendations, Assessment, Development and Evaluations (GRADE) approach to interpret the findings and rate the certainty of evidence [11], grading the major outcomes (mortality, clinical failure, and bacteremia development). Certainty of evidence was evaluated using the GRADEpro guideline (accessed on 6 May 2024) development tool software (GRADEpro GDT, Evidence Prime Inc., Hamilton, ON, Canada) [12], using parameters such as study design, risk of bias, directness of outcomes, heterogeneity, precision within results, bias due to publication, estimate effect, and dose relationship with response and confounders. The GRADE analysis rated the certainty of evidence as follows: high, moderate, low, or very low. We took this analysis into account in our conclusions.

## 3. Results

From 2012 to 2023, our search yielded 1849 citations, and 1844 records were excluded by two reviewers. Regarding M. Paul, et al. [5], 58 studies not using non-anti *Pseudomonas aeruginosa* antimicrobials were included. Finally, we identified 53 studies that included 9285 participants for the analysis of treatment failure [13,14,15,16,17,18,19,20,21,22,23,24,25,26,27,28,29,30,31,32,33,34,35,36,37,38,39,40,41,42,43,44,45,46,47,48,49,50,51,52,53,54,55,56,57,58,59,60,61,62,63,64,65,66,67,68,69,70,71,72,73,74,75,76] (Figure 1).

We summarized the year of publication, country, and type of antimicrobials in all the studies in Table 1. In our systematic review, the year of publication was mainly from 1990 to 2009 (72%), and the countries of publication were Turkey (*n* = 7), the USA (*n* = 6), Germany (*n* = 4), Japan (*n* = 3), the UK, Australia, Belgium, and the Netherlands (*n* = 2 each). The following beta-lactams covering *Pseudomonas aeruginosa* were used as monotherapy: carbapenems, ceftazidime (CAZ), CFPM, and PIPC/TAZ; which were used in 24, 12, 8, and 6 studies, respectively. In the combination therapy trials, CAZ, PIPC, PIPC/TAZ, CFPM, and cefoperazone/sulbactam (CPZ/SBT) were administered in 24, 9, 6, 4, and 4 studies, respectively. The same beta-lactam in both arms was compared in 16 trials, but 37 trials compared broad-spectrum beta-lactams such as carbapenem, beta-lactam/beta-lactamase inhibitor, and cefepime with narrower-spectrum beta-lactams such as CAZ (which insufficiently covers gram-positive bacteria) and PIPC plus aminoglycoside. For example, in the group of studies that compared different types of beta-lactams with an aminoglycoside, CAZ was used as monotherapy and in combination in 5 and 18 studies, respectively, and PIPC, azlocillin (AZL), carbenicillin (CAR), and ticarcillin (TIPC, which insufficiently cover anaerobes) were used as monotherapy and in combination in 1 and 13 studies, respectively. The characteristics of each study are shown in Appendix A.

### 3.1. Selection Bias

Funnel plot analyses were performed for the two main comparisons: all-cause mortality and treatment failure with any type of beta-lactam. The funnel plot for all-cause mortality was symmetrical (Appendix A). One potential indication within trials of treatment failure could be the absence of smaller trials supporting combination therapy (Appendix A).

### 3.2. Risk of Bias Assessment, GRADE Analysis, and Meta-Analysis

The risk of bias assessment data is presented in Figure 2. All-cause mortality in the same beta-lactam, different beta-lactam, and any type of beta-lactam was assessed, showing a risk ratio (RR) [95% CI] of 0.74 [0.53, 1.06] (heterogeneity: χ^2^ = 5.29; *p* = 0.81; I^2^ = 0%) in 10 RCTs, an RR [95% CI] of 0.97 [0.81, 1.16] (heterogeneity: χ^2^ = 14.92; *p* = 0.92; I^2^ = 0%) in 28 RCTs, and an RR [95% CI] of 0.99 [0.84, 1.16] (heterogeneity: χ^2^ = 33.57; *p* = 0.49; I^2^ = 0%) in 38 RCTs (Figure 3). Infection-related mortality in same beta-lactam, different beta-lactam, and any type of beta-lactam was assessed, showing an RR [95% CI] of 0.67 [0.42, 1.07] (heterogeneity: χ^2^ = 4.79; *p* = 0.69; I^2^ = 0%) in eight RCTs, an RR [95% CI] of 0.86 [0.67, 1.11] (heterogeneity: χ^2^ = 11.36; *p* = 0.91; I^2^ = 0%) in 26 RCTs, and an RR [95% CI] of 0.83 [0.66, 1.05] (heterogeneity: χ^2^ = 16.61; *p* = 0.94; I^2^ = 0%) in 34 RCTs (Figure 4). Treatment failure in same beta-lactam, different beta-lactam, and any type of beta-lactam was assessed, showing an RR [95% CI] of 1.11 [1.02, 1.20] (heterogeneity: χ^2^ = 17.31; *p* = 0.30; I^2^ = 13%) in 16 RCTs, an RR [95% CI] of 0.94 [0.89, 0.99] (heterogeneity: χ^2^ = 34.25; *p* = 0.55; I^2^ = 0%) in 37 RCTs, and an RR [95% CI] of 0.99 [0.94, 1.03] (heterogeneity: χ^2^ = 60.24; *p* = 0.20; I^2^ = 14%) in 53 RCTs (Figure 5). Renal failure as an adverse event was evaluated in two regimens: AG administered once daily or three times daily, and any dosing regimen. The RR [95% CI] was 0.26 [0.12, 0.53] (heterogeneity: χ^2^ = 0.95; *p* = 0.92; I^2^ = 0%) in six RTCs, was 0.51 [0.39, 0.68] (heterogeneity: χ^2^ = 12.65; *p* = 0.63; I^2^ = 0%) in 22 RCTs, and was 0.46 [0.36, 0.60] (heterogeneity: χ^2^ = 16.77; *p* = 0.67; I^2^ = 0%) in 28 RCTs (Figure 6). The duration of aminoglycoside treatment, the conduct of serum aminoglycoside monitoring, and the definition of renal failure differed among the various RCTs (Appendix A).

The GRADE analysis for all-cause and infection-related mortality with any of the beta-lactams used in the RCTs indicated “high certainty of evidence.” Moreover, the analysis of treatment failure with any of the beta-lactams used in the RCTs showed “moderate certainty of evidence”, and the analysis of renal failure with any dosing regimen revealed “high certainty of evidence” (Table 2).

### 3.3. Sensitivity Analysis

We performed the sensitivity analysis according to the age groups and publication year in Appendix A.

In the analysis of all-cause mortality for the different beta-lactam arm, newer publication years showed a trend favoring the combination of beta-lactam with aminoglycosides. For the 28 RCTs published between 1983–2019, the relative risk (RR) was 0.97 with a 95% confidence interval (CI) of 0.81–1.16 and an I^2^ of 0%, compared to an RR of 1.20 with a 95% CI of 0.83–1.74 and an I^2^ of 0% for the RCTs published between 2000 and 2019 in Appendix A.

In the analysis of infection-related mortality associated with the same beta-lactam, newer publication years showed a tendency to favor monotherapy. For eight randomized controlled trials (RCTs) published between 1983 and 2019, the relative risk (RR) was 0.67 with a 95% confidence interval (CI) of 0.42–1.07 and an I^2^ of 0%. This shifted to an RR of 0.53 with a 95% CI of 0.28–1.00 and an I^2^ of 0% for the four RCTs published between 2000 and 2019 shown in Appendix A.

For treatment failure, recent studies conducted between 2000 and 2019 in children showed that aminoglycosides did not add any benefits compared to monotherapy (7 RCTs, RR 0.77 [95% CI, 0.62–0.95], I^2^ 20%) with different beta-lactams. Conversely, in adult studies published between 2010 and 2019, aminoglycosides combined with the same beta-lactam regimen provided clear benefits (RR 1.10 [95% CI, 1.01–1.19], I^2^ 1%), as shown in Appendix A.

In examining nephrotoxicity across any aminoglycoside regimen, as shown in Appendix A, out of 28 randomized controlled trials (RCTs), only 11 reported monitoring serum aminoglycoside levels. There was no change in the relative risk (RR) observed. However, none of the publications reported the actual results of the aminoglycoside serum levels.

Review authors’ assessments of each risk of bias item is presented as percentages across all studies included in the review.

## 4. Discussion

In this systematic review, for the first time in 10 years, we extracted 53 trials that included 9407 participants and analyzed them to compare the efficacy of beta-lactam monotherapy with beta-lactam plus aminoglycoside combination therapy for empirically treating febrile neutropenic cancer patients. Although the details of the cancers in several studies are unknown, most of the participants included in these trials may have had hematological cancer such as leukemia and lymphoma. We assessed all-cause mortality, infection-related mortality, treatment failure (the definitions of these parameters were different in each study, as shown in Appendix A), and renal failure. Beta-lactam monotherapy might present a safer option than beta-lactam plus aminoglycoside combination therapy concerning the occurrence of adverse events such as renal failure. However, there were no significant differences observed regarding all-cause mortality and infection-related mortality between the combination therapy and monotherapy groups. Notably, the rate of treatment failure in combination therapy was significantly better for the same beta-lactam group, but in combination therapy was worse for the different beta-lactam group. We also performed a sensitivity analysis according to the publication year and patient age. The results from between 2010 and 2019 were better for the same beta-lactam groups and not significantly different for the different groups in adults. These findings reflect small but statistically significant changes in the outcomes.

In this study, the combination of aminoglycosides with narrow-spectrum beta-lactam antibiotics failed to provide the extensive coverage offered by broad-spectrum antibiotics. Cytotoxic chemotherapy-induced febrile neutropenia can lead to anaerobic bacteremia [1], as it makes the gastrointestinal mucosa vulnerable to infection from resident anaerobic bacteria. Compromised immunity and granulocytopenia enhance the risk of bacterial translocation and subsequent bloodstream infections [77]. Therefore, initial treatments typically include anti-anaerobic agents, especially when patients exhibit GI symptoms such as abdominal pain, diarrhea, or perianal discomfort. PIPC, AZL, CAR, or TIPC is not effective against the anaerobes. Furthermore, patients with hematological malignancies frequently required central venous catheterization. These patients are at high risk of infection by gram-positive strains such as *Staphylococcus aureus*, which are not fully covered by the beta-lactam and CAZ regimen. Accordingly, we infer that the combination therapy groups, when compared to the monotherapy groups, showed an increased rate of treatment failures, exemplified by extended fever or bacteremia from gram-positive cocci, prompting medical professionals to add vancomycin or carbapenems as alterations to the therapeutic strategy. Under the current guidelines for febrile neutropenia [1,2], neutropenic patients with hypotension and persistent fever without identifiable sources should broaden their antimicrobial regimen to cover drug-resistant gram-negative and gram-positive organisms as well as anaerobes. Initial regimens should include carbapenems, CFPM, or PIPC/TAZ combined with an aminoglycoside or vancomycin. However, CAZ, PIPC, AZL, CAR, or TIPC is not recommended.

Another reason could be that most RCTs included small numbers of resistant strains of gram-negative bacilli, were published between 1990 and 2009, and excluded the pre-treatment of infections caused by strains resistant to the study drugs (Appendix A). However, the burden of resistant strains these days is higher than in the past [78]. Therefore, we have a potential increase in resistant bacteria among gram-negative bacilli, which were rarely detected, this time, in this systematic review. Infections caused by multidrug-resistant (MDR) gram-negative bacilli, including those caused by extended-spectrum beta-lactamase (ESBL)-producing Enterobacterales and carbapenem-resistant gram-negative bacilli, are increasing worldwide among cancer patients [79]. The treatment of MDR gram-negative bacilli infections in patients without neutropenia involves combination therapy of aminoglycoside [80,81,82,83]. For example, in the AMINOLACTAM study [7], the combination therapy group showed a lower seven-day case fatality rate. This was because the most detected pathogen was *E. coli* (50.4%), and the rate of MDR bloodstream infection (BSI) was 27.7%. These rates are higher than those in the 53 studies in our systematic review. Therefore, combination therapy of aminoglycoside should be considered in the setting of highly resistant strains.

In our systematic review, aminoglycoside combination therapy groups showed higher nephrotoxicity than monotherapy groups. In the results of the AMINOLACTAM study [7], the use of aminoglycosides did not show a significant association with impairment of renal function. In the study, the median duration of aminoglycoside treatment was two days. In general, patients with febrile neutropenia developed kidney failure due to sepsis or vancomycin. In a previous study [84], kidney failure in patients with febrile neutropenia was related to sepsis and not aminoglycoside. In our systematic review, we observed that a significant number of the included studies failed to conduct serum aminoglycoside monitoring (Appendix A). Furthermore, a sensitivity analysis was conducted on 11 RCTs monitoring serum aminoglycoside, but the results of the risk ratio for nephrotoxicity did not change. This could be because the majority of studies administered the aminoglycoside for a longer period than the AMINOLACTAM study did, or because there were no articles reporting the results of aminoglycoside concentration levels. Even if they were measured, it is possible that there was nephrotoxicity or efficacy against the infection, but many of the studies were old and the details could not be investigated. Esteve (1997) [30] and de Pauw (1994) [85] observed a strong impact of aminoglycoside on the rate of renal failure. However, in the study by de Pauw (1994), only 44% of patients required dose adjustment to ensure that serum aminoglycoside levels were within the therapeutic range. In the study by Esteve (1997), 33% of patients undergoing combination therapy were also administered vancomycin. We have to prevent renal failure due to aminoglycoside administration by performing therapeutic drug monitoring and considering combination therapy for short durations.

On the other hand, with treatment guidelines having been updated over the past 10 years, treatment options other than aminoglycosides have also emerged. As therapeutic strategies for gram-negative resistant bacteria, we will discuss (1) prolonged infusion of beta-lactams for bacteria with high minimum inhibitory concentration (MIC), (2) combination therapy with non-beta-lactam antibiotics, such as tigecycline and colistin, that include aminoglycosides, and (3) new beta-lactam/beta-lactamase inhibitors.

### 4.1. Prolonged Infusion Beta-Lactams

Beta-lactam antibiotics are typically administered as a bolus over 30 min. Prolonged infusion administration strategies for intravenous beta-lactam antibiotics may include either an extended infusion (over 2 to 4 h) or a continuous infusion (over the entire dosing interval).

With prolonged infusion, after loading with a bolus dose such as 30 min, the subsequent doses are given over an extended period that is half the usual administration time. According to PK-PD theory, this approach extends the duration in which the drug concentration exceeds the MIC (Time above MIC) [86]. This allows for treatment even when the MIC is high, and the susceptibility is intermediate. Further research is anticipated on prolonged infusion targeting high MIC in patients with hematologic malignancies. Moreover, prolonged infusion has shown additional efficacy beyond targeting high MIC strains in patients with severe conditions.

In sepsis, the intravascular concentration of the antibiotic decreases as the drug moves to extravascular spaces like the third space, leading to reduced clinical effectiveness. Therefore, the guidelines recommend prolonged infusion in sepsis or septic shock [87]. One RCT including patients with febrile neutropenia suggests that the prolonged or continuous infusion of beta-lactam has shown greater clinical improvement in patients [88]. In the study on pediatric febrile neutropenia in Mexico, the targeted patients had hematological malignancies [89]. Continuous infusion showed a trend toward clinical improvement, although it did not reach statistical significance.

From the above perspectives, prolonged infusion of beta-lactam for febrile neutropenia may be a candidate alone as well as in combination with aminoglycoside.

### 4.2. Combination Therapy

In antimicrobial combination therapy, there have been clinical studies conducted in Greece and Italy that targeted bloodstream infections caused by carbapenemase, using agents other than beta-lactam antibiotics [82,90]. Reports indicate that combination therapies, including carbapenems, have lower mortality rates than therapies excluding carbapenems or monotherapies, and they are frequently used in clinical practice. Combination options with beta-lactam antibiotics include tigecycline or colistin other than aminoglycoside.

### 4.3. Tigecycline

Although tigecycline is ineffective against *Proteus vulgaris, Pseudomonas aeruginosa*, and *Providencia* sp., it can be used for CRE and carbapenem-producing Enterobacteriaceae (CPE). Tigecycline has in vitro activity against methicillin-resistant *Staphylococcus aureus*, anaerobic bacteria, ESBL-producing bacteria, and intracellular pathogens. However, tigecycline is bacteriostatic and has a large volume of distribution; care should be taken as it transitions to the third space. There have been reports of high mortality rates in hospital-acquired pneumonia, including ventilator-associated pneumonia, and intra-abdominal infections when used as monotherapy. Therefore, it should always be used in combination therapy [91,92]. Side effects include hepatic dysfunction and gastrointestinal symptoms. While there are some studies on the use of tigecycline in hematologic malignancy patients, almost none have identified resistance genes. Some reports suggest that susceptibility to tigecycline is favorable in carbapenem strains with a higher MIC [93,94,95], but the clinical outcomes are not consistent [96,97,98]. One study indicates that combining piperacillin/tazobactam with tigecycline is considered safe and more effective compared to using piperacillin/tazobactam by itself in febrile neutropenic hematologic cancer patients at high risk [99]. Unlike aminoglycosides, tigecycline does not cover *Pseudomonas aeruginosa*, so for febrile neutropenia, it might be prudent to consider it as definitive therapy only after the pathogen has been identified.

### 4.4. Colistin

Another option for combination therapy is colistin, a polypeptide antibiotic. It exhibits antibacterial activity by binding to the bacterial outer membrane. It shows effectiveness against *Pseudomonas aeruginosa, Acinetobacter* sp., *Escherichia coli, Klebsiella* sp., *Enterobacter* sp., and *Citrobacter* sp. However, it is ineffective against gram-positive bacteria, *Burkholderia* sp., *Neisseria* sp., *Proteus* sp., *Serratia* sp., *Providencia* sp., *Brucella* sp., and anaerobes. Loading is required when using colistin [100], and it should not be used as monotherapy. Colistin methanesulfonate (CMS) is intravenously administered, where 30% of it is hydrolyzed into the active antibacterial form of colistin. The remaining 70%, which lacks antibacterial activity, is excreted via the kidneys as CMS. Of the hydrolyzed colistin in active form, 70% is excreted outside the kidneys, with the remaining 30% excreted in the urine. Consequently, of the administered CMS, only about 9% of it transitions to colistin in the urine, making it less suitable for treating urinary tract infections [101]. The main side effects of colistin injection are renal impairment and neurological damage. Studies on the use of colistin in hematologic malignancy patients are limited. However, there are reports indicating its efficacy and susceptibility in CRE hematological malignancy patients [102,103], and others that highlight good tolerability of colistin in terms of neurotoxicity and nephrotoxicity [104] in hematological malignancy patients. In hematological malignancy patients with CRE, when considering combination therapy with beta-lactams, whether to use aminoglycosides or colistin should be determined in conjunction with the facility’s antibiogram.

### 4.5. Novel Beta-Lactam/Beta-Lactamase Inhibitors

According to Ambler’s classification that categorizes beta-lactamases [105], beta-lactamases are divided into Classes A–D, with each class acting differently on distinct beta-lactam lineages (Figure 7). Additionally, Classes A, C, and D comprise a type of serine beta-lactamase with a serine residue at their active center, whereas Class B is a type of metallo beta-lactamase containing zinc. In effective beta-lactam/beta-lactamase inhibitor, for Class A carbapenemases like *Klebsiella pneumoniae* carbapenemase (KPC), Imipenem/cilastatin/relebactam and ceftazidime/avibactam (CAZ/AVI) are effective. For class B, IDSA Guidelines recommend combined therapy of ceftazidime-avibactam and aztreonam [106]. The novel antibiotic cefiderocol is not degraded by beta-lactamase and is effective against Class A and Class B. It remains effective even with a decrease in porins that allows the uptake of antibiotics into bacteria and is unaffected by efflux pumps that expel antibiotics from within the bacteria. This is why not only is cefiderocol effective against class A, B, and D carbapenemases, but it is also effective against *Pseudomonas aeruginosa* and carbapenem-resistant *Acinetobacter* sp. [107]. Additionally, while Ceftolozane/Tazobactam is ineffective against CPE, it has shown efficacy against ESBL-producing bacteria, AmpC-producing bacteria, and CRE [108]. In empirical therapy for febrile neutropenia patients, its safety and efficacy have been demonstrated in RCT and cohort studies when compared with CFPM, PIPC/TAZ, and meropenem. It is anticipated to be a promising empirical carbapenem-sparing regimen for the patients with hematological malignancy in febrile neutropenia in the future [109,110].

One retrospective study shows that CAZ/AVI may be considered the preferred therapeutic option for KPC-producing Enterobacteriaceae blood stream infections in high-risk neutropenic patients [111].

The antibiotics introduced in this session are summarized in Table 3.

There are various treatment options besides the combined therapy of aminoglycosides and beta-lactams. It is essential to choose appropriate antimicrobials in accordance with the local antibiogram and the Antimicrobial Stewardship Program.

## 5. Conclusions

In our systematic review, beta-lactam monotherapy is safer compared with combination therapy concerning adverse events. Treatment failure may be lower in the combination therapy of the same beta-lactam group. However, a narrower spectrum of beta-lactams was included in the combination therapy of different groups, and almost all RCTs did not mention the resistant strains of gram-negative bacilli.

We could not reduce the use of broad-spectrum antibiotics by combining aminoglycosides with a narrow-spectrum beta-lactam. On the other hand, we were also unable to conduct a detailed investigation of serum aminoglycoside levels. In the future, studies that include the profiles of antibiotic-resistant bacteria and the monitoring of serum aminoglycoside levels will be required.

## Figures and Tables

**Figure 1 cancers-16-01934-f001:**
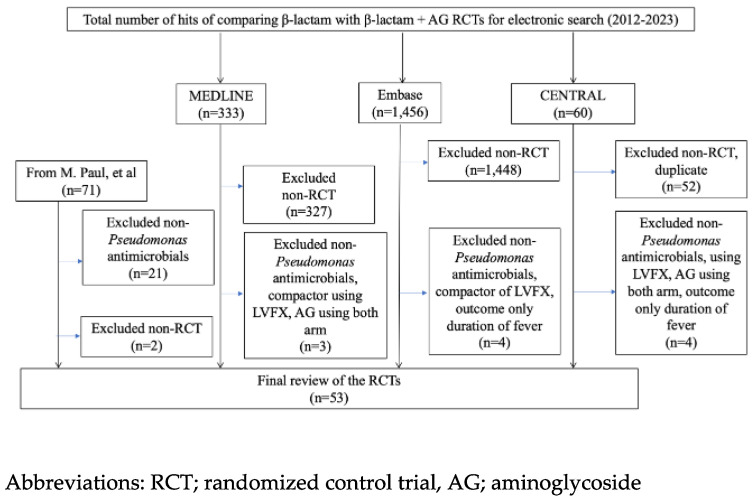
Identification process for eligible studies.

**Figure 2 cancers-16-01934-f002:**
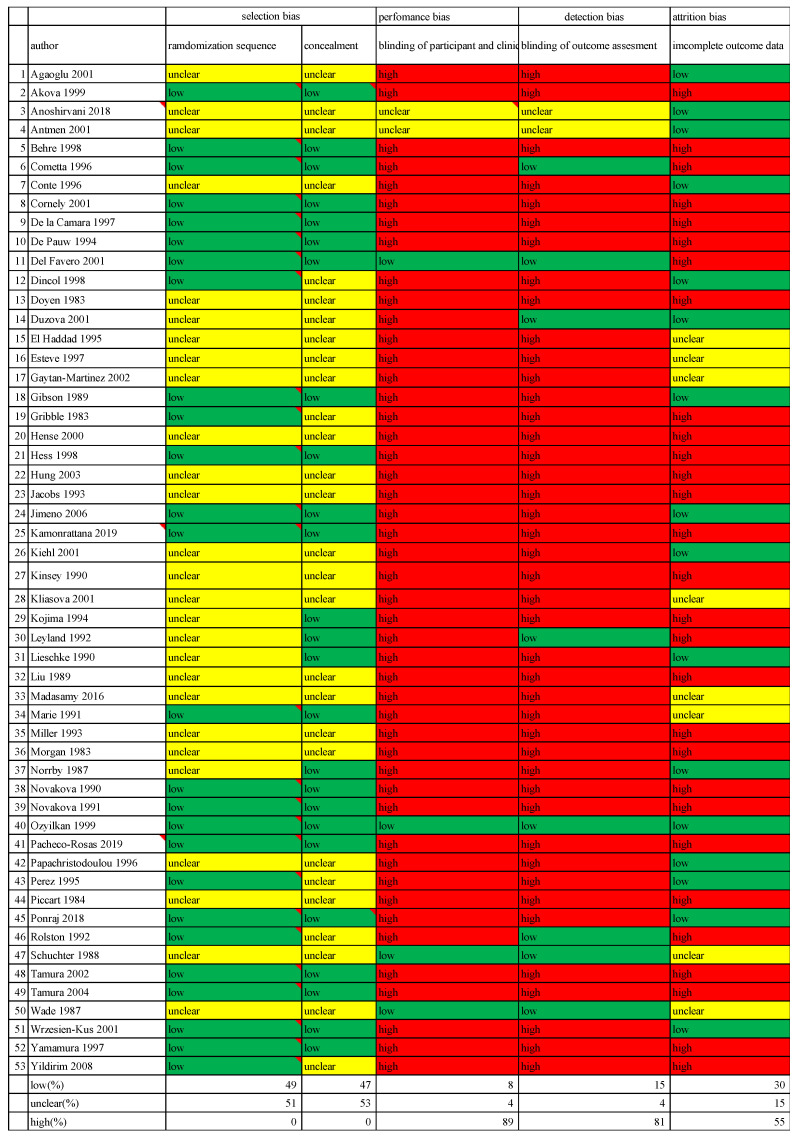
Risk of bias graph [13,14,15,16,17,18,19,20,21,22,23,24,25,26,27,28,29,30,31,32,33,34,35,36,37,38,39,40,41,42,43,44,45,46,47,48,49,50,51,52,53,54,55,56,57,58,59,60,61,62,63,64,65,66,67,68,69,70,71,72,73,74,75,76].

**Figure 3 cancers-16-01934-f003:**
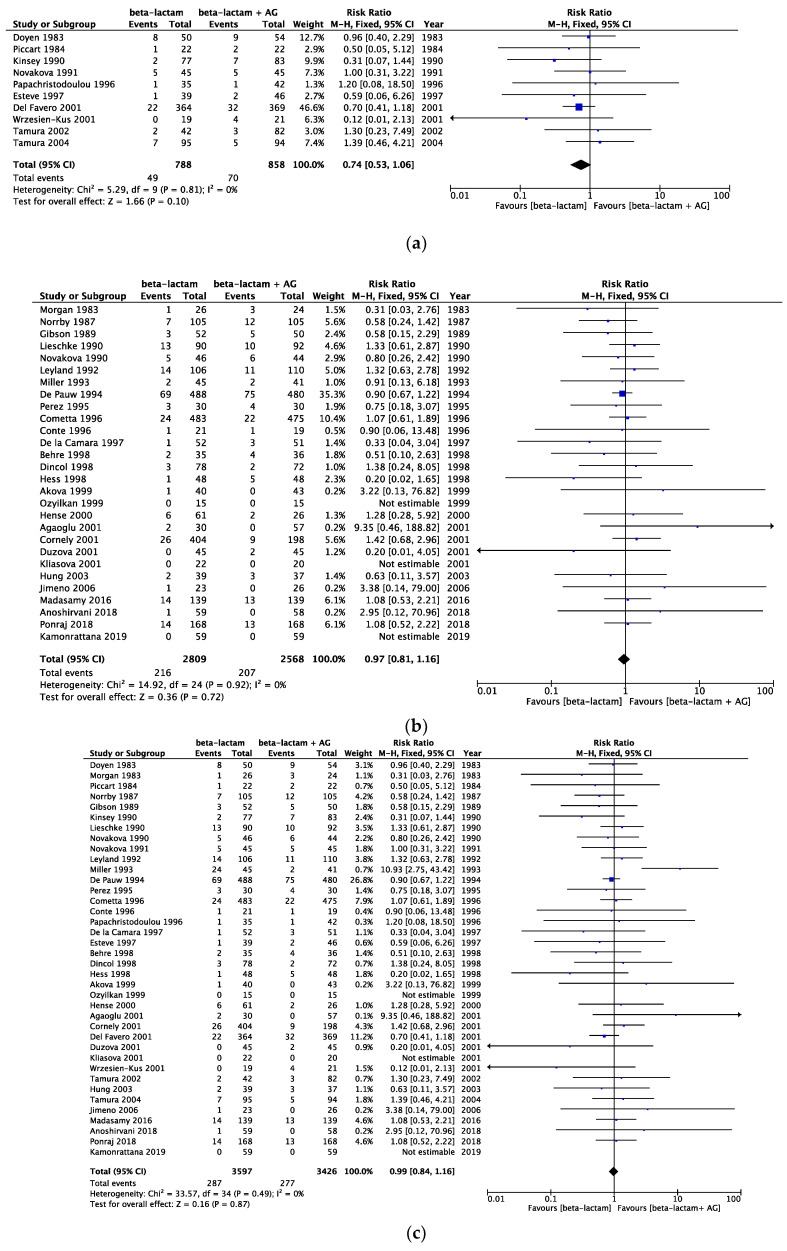
(**a**) Comparison of all-cause mortality between monotherapy and AG combination therapy in same beta-lactam. Abbreviations: CI; confidence interval, AG; aminoglycoside. (**b**) Comparison of all-cause mortality between monotherapy and AG combination therapy in different beta-lactam. Abbreviations: CI; confidence interval, AG; aminoglycoside. (**c**) Comparison of all-cause mortality between monotherapy and AG combination therapy in all beta-lactam. Abbreviations: CI; confidence interval, AG; aminoglycoside [13,14,15,16,17,18,19,20,21,22,23,24,25,26,27,28,29,30,31,32,33,34,35,36,37,38,39,40,41,42,43,44,45,46,47,48,49,50,51,52,53,54,55,56,57,58,59,60,61,62,63,64,65,66,67,68,69,70,71,72,73,74,75,76].

**Figure 4 cancers-16-01934-f004:**
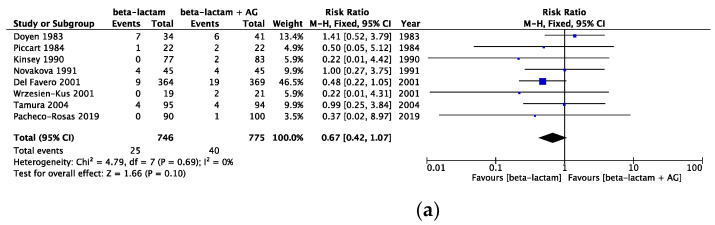
(**a**) Comparison of infection-related mortality between monotherapy and AG combination therapy in same beta-lactam. Abbreviations: CI; confidence interval, AG; aminoglycoside. (**b**) Comparison of infection-related between monotherapy and AG combination therapy in different beta-lactam. Abbreviations: CI; confidence interval, AG; aminoglycoside. (**c**) Comparison of infection-related between monotherapy and AG combination therapy in all beta-lactam. Abbreviations: CI; confidence interval, AG; aminoglycoside [13,14,15,16,17,18,19,20,21,22,23,24,25,26,27,28,29,30,31,32,33,34,35,36,37,38,39,40,41,42,43,44,45,46,47,48,49,50,51,52,53,54,55,56,57,58,59,60,61,62,63,64,65,66,67,68,69,70,71,72,73,74,75,76].

**Figure 5 cancers-16-01934-f005:**
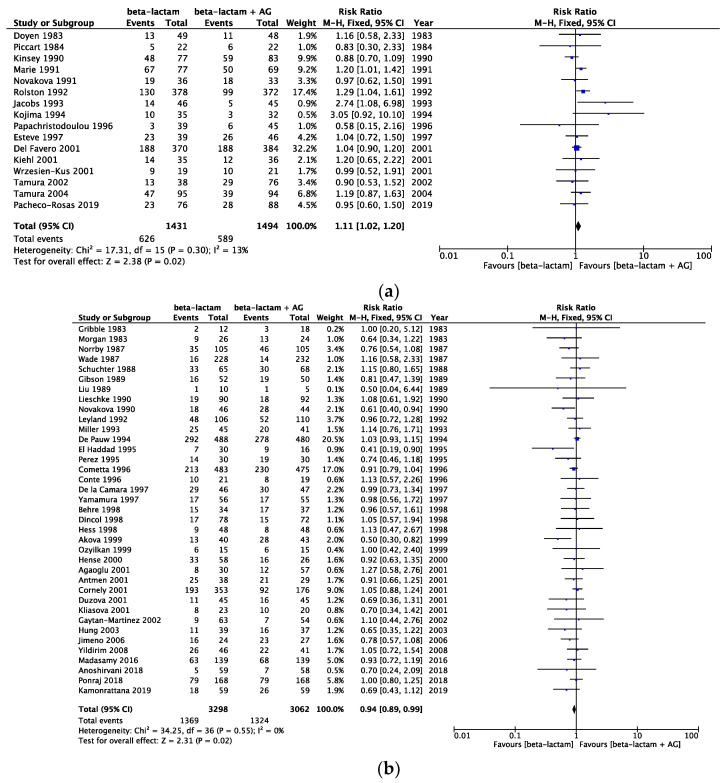
(**a**) Comparison of treatment failure between monotherapy and AG combination therapy in same beta-lactam. Abbreviations: CI; confidence interval, AG; aminoglycoside. (**b**) Comparison of treatment failure between monotherapy and AG combination therapy in different beta-lactam. Abbreviations: CI; confidence interval, AG; aminoglycoside. (**c**) Comparison of treatment failure between monotherapy and AG combination therapy in all beta-lactam. Abbreviations: CI; confidence interval, AG; aminoglycoside [13,14,15,16,17,18,19,20,21,22,23,24,25,26,27,28,29,30,31,32,33,34,35,36,37,38,39,40,41,42,43,44,45,46,47,48,49,50,51,52,53,54,55,56,57,58,59,60,61,62,63,64,65,66,67,68,69,70,71,72,73,74,75,76].

**Figure 6 cancers-16-01934-f006:**
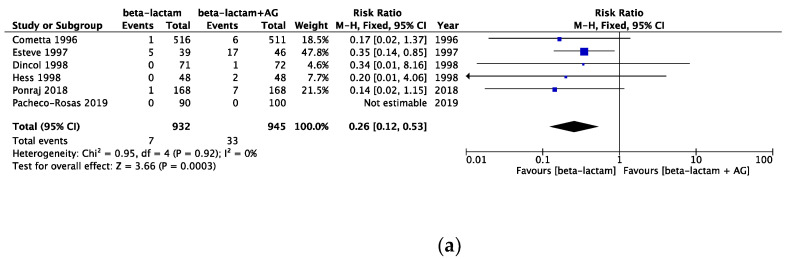
(**a**) Comparison of nephrotoxicity between monotherapy and AG combination therapy in the once daily regimen. Abbreviations: CI; confidence interval, AG; aminoglycoside. (**b**) Comparison of nephrotoxicity between monotherapy versus AG combination therapy in the multiple daily dosing regimen. Abbreviations: CI; confidence interval, AG; aminoglycoside. (**c**) Comparison of nephrotoxicity between monotherapy and AG combination therapy in any daily regimen. Abbreviations: CI; confidence interval, AG; aminoglycoside [13,14,15,16,17,18,19,20,21,22,23,24,25,26,27,28,29,30,31,32,33,34,35,36,37,38,39,40,41,42,43,44,45,46,47,48,49,50,51,52,53,54,55,56,57,58,59,60,61,62,63,64,65,66,67,68,69,70,71,72,73,74,75,76].

**Figure 7 cancers-16-01934-f007:**
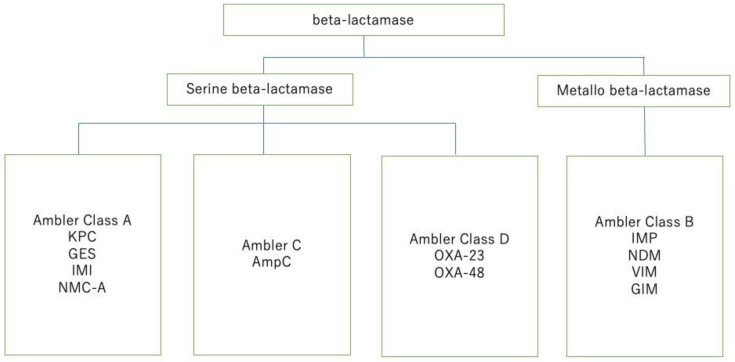
Ambler classification.

**Table 1 cancers-16-01934-t001:** The characteristics of the 53 RCTs.

Published Year		Published Country		Type of Antimicrobial in Beta-Lactam Monotherapy Arm		Type of Antimicrobial in Beta-Lactam + AG Arm		Type of Beta-Lactam between Same Beta-Lactam and Beta-Lactam + AG (16)			
1983–1989	10	Turkey	7	MEPM	11	IMP/CS + AG	1	Carbapenem	1	CAZ or Carbapenem	1
1990–1999	23	USA	6	IMP/CS	11	CFPM + AG	4	beta-lactam/beta-lactamase inhibitor	4	CPZ	1
2000–2009	15	Germany	4	IMP/CS or MEPM	1	CPZ/SBT + AG	4	CFPM	3		
2010–2019	5	Japan	3	PIPC/TAZ	6	CAZ + AG	23	CAZ	6		
		UK, Australia, Belgium, Netherlands, Mexico, Chile, India	2	CAZ	11	CAZ or IMP/CS + AG	1	type of beta-lactam between different beta-lactam and beta-lactam + AG (37)			
		other	19	CFPM	8	carbenicillin + AG	1	Carbapenem	22	beta-lactam/ beta-lactamase inhibitor + AG	6
				CAZ or IMP/CS	1	azlocillin + AG	2	beta-lactam/ beta-lactamase inhibitor	4	CAZ + AG	17
				CPZ/SBT	2	CPZ + AG	1	CFPM	5	CAZ + AG or CFPM + AG	1
				CPZ	1	PIPC/TAZ + AG	6	CAZ	5	PIPC/AZL/CAR/TIPC + AG	13
				PIPC	1	PIPC + AG	9	PIPC/AZL/CAR/TI‘C	1		
						ticarcillin + AG	1				

Abbreviations: MEPM, meropenem; IMP/CS, imipenem/cilastatin; PIPC/TAZ, piperacillin/tazobactam; CAZ, ceftazidime; CFPM, cefepime; CPZ/SBT, cefoperazone/sulbactam; AG, aminoglycoside.

**Table 2 cancers-16-01934-t002:** Summary of findings regarding the administration of beta-lactam compared to beta-lactam plus aminoglycoside for febrile neutropenia.

Beta-Lactam Compared to Beta-Lactam + Aminoglycoside for Febrile Neutropenia						
Patient or population: febrile neutropenia; Intervention: beta-lactam; Comparison: beta-lactam + aminoglycoside						
Outcomes	Anticipated absolute effects * (95% CI)		Relative effect (95% CI)	№ of participants (studies)	Certainty of the evidence (GRADE)	Comments
	Risk with beta-lactam + aminoglycoside	Risk with beta-lactam				
All-cause mortality	81 per 1000	80 per 1000 (68 to 94)	OR 0.99 (0.84 to 1.16)	7023 (38 RCTs)	⨁⨁⨁⨁ High	
Infection-related mortality	50 per 1000	42 per 1000 (33 to 53)	OR 0.83 (0.66 to 1.05)	6469 (34 RCTs)	⨁⨁⨁⨁ High	
Treatment failure	420 per 1000	416 per 1000 (395 to 432)	OR 0.99 (0.94 to 1.03)	9285 (53 RCTs)	⨁⨁◯◯ Moderate ^a,b^	
Ag any dosing regimen	54 per 1000	25 per 1000 (19 to 32)	RR 0.46 (0.36 to 0.60)	6007 (28 RCTs)	⨁⨁⨁⨁ High	

* The risk in the intervention group (and its 95% confidence interval) is based on the assumed risk in the comparison group and the relative effect of the intervention (and its 95% CI). CI: confidence interval; OR: odds ratio; RR: risk ratio; Ag: aminoglycoside; a. Differences decreased with low risk of bias regarding allocation concealment. b. Differences in effects between published and unpublished trials. GRADE Working Group grades of evidence. High certainty: we are very confident that the true effect lies close to the estimated effect. Moderate certainty: we are moderately confident in the estimated effect: the true effect is likely to be close to the estimated effect, but there is a possibility that it is substantially different. Low certainty: our confidence in the estimated effect is limited: the true effect may be substantially different from the estimated effect. Very low certainty: we have very little confidence in the estimated effect: the true effect is likely to be substantially different from the estimated effect.

**Table 3 cancers-16-01934-t003:** Summary of antibiotics for gram-negative bacilli.

Antibiotics	CPE			CRE	ESBL	AmpC	MRDP
	KPC	MBL	OXA-48				
Cefepime	× *	× *	× *	× *	×	◯	× *
Piperacillin/tazobactam	× *	× *	× *	× *	△	△	× *
Aztreonam	× *	◯	× *	×	×	×	× *
Ceftazidime	× *	× *	× *	× *	×	×	× *
Carbapenem	× *	× *	× *	× *	◯	◯	× *
Ceftolozane/tazobactam	× *	× *	× *	◯	◯	◯	◯
Imipenem/Cilastatin/ Relebactam	◯	×	×	◯	◯	◯	◯
Cefiderocol	◯	◯	◯	◯	◯	◯	◯
** Colistin	◯	◯	◯	◯	◯	◯	◯
*** Tigecycline	◯	◯	◯	◯	◯	◯	×

* If administered over an extended period, efficacy may be observed even at MICs within the intermediate range. ** Colistin is administered as a prodrug. It is not recommended for monotherapy, hence it is often combined with beta-lactam antibiotics such as carbapenems. Its renal transition is poor, and it has potential neurotoxic and nephrotoxic effects. *** Tigecycline is not recommended for monotherapy, hence it is often combined with beta-lactam antibiotics such as carbapenems. It has bacteriostatic properties and is less preferred for bacteremia. Abbreviation; CPE, Carbapenemase-Producing Enterobacteriaceae; CRE, Carbapenem-Resistant, Enterobacteriaceae; ESBL, Extended-Spectrum Beta-Lactamase; AmpC, AmpC Beta-Lactamase; MRDP, multi-drug resistant *Pseudomonas aeruginosa*; KPC, Klebsiella pneumoniae Carbapenemase; MBL, Metallo-Beta-Lactamase; OXA-48, Oxacillinase-48.

## Data Availability

The data presented in this study are available on request from the corresponding author (Kazuhiro Ishikawa).

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
