# Peer review of "Systematic Review of Beta-Lactam vs. Beta-Lactam plus Aminoglycoside Combination Therapy in Neutropenic Cancer Patients"

_cancers, 2024, doi:10.3390/cancers16101934_

Round 1

Reviewer 1 Report (Previous Reviewer 2)

Comments and Suggestions for Authors

Dear Authors

Thank you for acept all of the revision comments, I belive that the results are now stronger that in the previous version. 

I only have a last considetarion  seen the new results added. In line 249-250 you mencioned that there is not a clear trend in table S5 in the sensivity analysis of treatment failure.  I do not agree with the autors, I see a relevant result, in recent studies after 2000 in children, the aminoglycoside did not add benefits compared with monotheraphy 7 RCTs RR 0.77 (95%CI, 0.62-0.95) I2 20%, on contray in adults it the last 9 years (probably with more accurate regimen), in adults with the same b-lactma, aminoglycose it's a clear benefits RR 1.10  (95%CI, 1.01-1.19) I2 1%.

I suggets add this results on abstract, results section and in discussion. 

Author Response

Thank you for your feedback. 

According to the feedback. 

I added the following sentence, Line 27-29, of the abstract.

"In the sensitivity analysis, the treatment failure data published between 2010 and 2019 show better outcomes in the same beta-lactam group (RR 1.10 [95% CI, 1.01-1.19]). "

in lines 249-254 of the result section. 

"For treatment failure, recent studies conducted between 2000 and 2019 in children show that aminoglycosides did not add any benefits compared to monotherapy (7 RCTs, RR 0.77 [95% CI, 0.62-0.95], I² 20%) with different beta-lactams. Conversely, in adults between 2010 and 2019, aminoglycosides combined with the same beta-lactam regimen provided clear benefits (RR 1.10 [95% CI, 1.01-1.19], I² 1%), as shown in Supplemental Table 5."

Also, in Lines 313-318 of the discussion part.

"Notably, treatment failure in combination therapy was significantly better in the same beta-lactam group, but that of combination therapy was worse in the different beta-lactam group. We also performed a sensitivity analysis according to the publication year and age. The results between 2010 and 2019 are better in the same beta-lactam groups and not significantly different in the different groups in adults."

Reviewer 2 Report (Previous Reviewer 1)

Comments and Suggestions for Authors

Thank you for this approved manuscript that I was happy to read.

Author Response

Thank you for your great feedback. 

This manuscript is a resubmission of an earlier submission. The following is a list of the peer review reports and author responses from that submission.

Round 1

Reviewer 1 Report

Comments and Suggestions for Authors

Thank you for the opportunity to review this paper by a well-established group of authors.

The paper is a meta analysis of papers comparing beta-lactam antibiotics as monotherapy vs beta-lactam antiboticcs with the addition of an aminoglycoside in patients with febrile neutropenia.

The patients treated in the studies in the metaanalysis includes patients with solid cancer as well as hematological malignancies – it is stated that “most” patients belonged to the latter category.

The review add to a previous Cochrane analysis from 2013

The overall conclusions of the paper are clear – upfront addition of aminoglycoside does not improve the infectioncontrol, moreover renal adverse events were more common in the aminoglycoside group.

The generalizability of the findings are hampered by

  • Different diseases: solid vs hematological malignancy, expected duration of neutropeni
  • Different antibiotic regimens
    • Some of these regimens will be used rarely at present
    • Sometimes different betalactams in the aminoglycoside arm compared to the normal arm
      • To accommodate for this the authors give different tables for the groups
    • No definition of number of aminoglycoside adminsitrations: single dose, three days, until culture negativity, and is it guided by serum measurements? (as stated in l 322 indeed the excess nephrotoxicity is most pronounced in old studies wo plasma-value controls.

The figures 3- 6. 

                             The studies are mentioned alphabetically. Why not by year  or size

                           There are no references stated – I find it would improve readability to have the references in the paper and not in the appendix

                             It would be helpful to state the used antibiotics for easier comparisons to local treatments

                             In some of the figures it is stated Favors beta lactam vs favors betalactam + AG which is preferred compared to “favors experimental vs control” used in some figures.

The section 2 on combination therapies l 356ff  regarding non-aminoglycosides as tigecycline, coli-stine 363-402 is some what beyond the scope of the review but I find it acceptable – it could be stated in the abstract that combination therapies with non aminoglycosides is discussed. Could the authors comment on the potential use of flourochinolones as alternative to AGs in some patients if relevant in their opinion.

L 67 “Spanish study”. The study was multinational. Please explain why the AMINOLACTAM study is noy included in the analyses presented here –

L 90  The authors are complimented for posting their review protocol prior to the study

Fig 2a Funnel plat analysis would it be relevant to name the outlier study. Could these plots be placed in the appendix?

L181 CFPM – first mention please writ  cefepime

L 230 Defintion of “renal failure”?

Fig 3b “favous beta lactam” please correct

Fig 3 c L 237 combination not ombination

Fig 4a Comment on the somewhat contraintuitove finding by del favro the largest study (pip/tazo vs pip/tazo + AG) In the discussion session.

Fig 4 title b l 237 comparion  of what?

4 c l 246 do

Fig 5a and 5b the findings of small albeit statistically significant changes in favor of betalaktamAG if the same betalactam was used and the opposite finding in the case of different betalactams presumably with a more narrow spectrum could be used in the discussion-

Table 2 – These data seems completely different from the data presented in the figures and the conclusions of the analysis???!!!

l 285 “we were unable to spare.:.” rephrase…

l 296-298 important point but please consider rephasing.

L 322- it is important that it is mentioned that the studies that drive the negative effect of AG on nephrotoxicity was without plasma controls.

L 338 ff “prolonged infusion” – interesting points. Many/some institutions now use 24 hour pump infusions of betalactams. The comments about prolonged infusion could perhaps be moved to novel beta-lactam section as optimized or novel….

L 379 ref nr 36 = wrong ref should be nr 35 (please check other refs as well)!!!!

L 447 treatment failure may be the favourable outcome.. rephrase

L 451 we were unable to spare… rephrase

L 513 ref 21??

Comments on the Quality of English Language

-

Author Response

(1) The figures 3- 6. 

(1)The studies are mentioned alphabetically. Why not by year  or size. There are no references stated – I find it would improve readability to have the references in the paper and not in the appendix.

Response> I have sorted the studies chronologically and included the references within the manuscript for improved readability.

It would be helpful to state the used antibiotics for easier comparisons to local treatments

Response> List the antimicrobials used and figure is written in a supplement table1.

In some of the figures it is stated Favors beta lactam vs favors betalactam + AG which is preferred compared to “favors experimental vs control” used in some figures.

Response> Thank you for your feedback.

(2)The section 2 on combination therapies l 356ff  regarding non-aminoglycosides as tigecycline, coli-stine 363-402 is some what beyond the scope of the review but I find it acceptable – it could be stated in the abstract that combination therapies with non aminoglycosides is discussed. Could the authors comment on the potential use of flourochinolones as alternative to AGs in some patients if relevant in their opinion.

Response> We need to be cautious when using fluoroquinolones for febrile neutropenia. This is because, as noted in guidelines, fluoroquinolones have been used as prophylaxis and shown by meta-analysis to reduce infections such as those caused by Gram-negative rods (GNR) in patients with neutropenia (PMID: 9508206). However, there are concerns about resistance (PMID: 37746769), leading to divided opinions. Therefore, in cases of febrile neutropenia, they should not be used in patients already exposed to fluoroquinolones due to the risk of antimicrobial resistance.

(3)L 67 “Spanish study”. The study was multinational. Please explain why the AMINOLACTAM study is noy included in the analyses presented here –

Response> In this meta-analysis, we included only randomized controlled trials (RCTs). AMINOLACTAM is a retrospective study; therefore, we excluded it from our analysis.

(4)L 90  The authors are complimented for posting their review protocol prior to the study

Response>Thank you for feedback.

(5)Fig 2a Funnel plat analysis would it be relevant to name the outlier study. Could these plots be placed in the appendix?

Response> As your feedback, I moved to the supplement figure 2a, 2b.

(6)L181 9 – first mention please writ  cefepime

Response> I also spell out cefepime (CFPM) in line 49.

(7)L 230 Defintion of “renal failure”?

Response>I have created a new supplement table 2 detailing whether the definition of renal failure was provided in each study. As you mentioned, not all studies included in our analysis provided a clear definition of renal failure, which is an important consideration. I added the definition of renal failure in Line 124-126 and Line 230-232.

(8)Fig 3b “favous beta lactam” please correct

Response> I corrected as favours.

(9)Fig 3 c L 237 combination not ombination

Response> I corrected as combination.

(10)Fig 4a Comment on the somewhat contraintuitove finding by del favro the largest study (pip/tazo vs pip/tazo + AG) In the discussion session.

Response> Figure 4a illustrates infection-related mortality for the same beta-lactam. The study by Del Favero in 2001 had the largest sample size among the studies included in Figure 4a. This study involved adults with leukemia (62%), lymphoma (19%), solid tumors (8%), and bone marrow transplant recipients (27%). The comparator arms were PIPC/TAZ versus PIPC/TAZ plus amikacin. The risk ratio for infection-related mortality was observed to be across 1 in both arms, indicating no significant difference between them. However, there was no information available regarding serum levels of aminoglycoside. The susceptibility tests for PIPC/TAZ and amikacin against gram-negative rods were almost similar. Therefore, these factors did not show any difference in infection-related mortality between PIPC/TAZ alone and PIPC/TAZ plus AMK.

(11)Fig 4 title b l 237 comparion  of what? (12)4 c l 246 do

Response> Fig 4b title  is “Comparison of infection-related between monotherapy and AG combination therapy in different beta-lactam.” and Fig 4c title is “Comparison of infection-related between monotherapy and AG combination therapy in all  beta-lactam.”

(13)Fig 5a and 5b the findings of small albeit statistically significant changes in favor of betalaktamAG if the same betalactam was used and the opposite finding in the case of different betalactams presumably with a more narrow spectrum could be used in the discussion-

Response> I added the “These findings reflect small but statistically significant changes in the outcomes.” In Line 332.

(14)Table 2 – These data seems completely different from the data presented in the figures and the conclusions of the analysis???!!!

Response> I have updated the GRADE assessment for treatment failure in line 235-236. The analysis now indicates "moderate certainty of evidence" for treatment failure with any of the beta-lactams used in the RCTs.

(15)l 285 “we were unable to spare.:.” rephrase…

Response> I have rephrased the statement as “the combination of aminoglycosides with narrow-spectrum beta-lactam antibiotics failed to provide the extensive coverage offered by broad-spectrum antibiotics.” In Line 332-334.

(16)l 296-298 important point but please consider rephasing.

Response> I have rephrased the statement as “Accordingly, we infer that the combination therapy groups, when compared to the monotherapy groups, showed an increased rate of treatment failures, exemplified by extended fever or bacteremia from gram-positive cocci, prompting medical professionals to add vancomycin or carbapenems as alterations to the therapeutic strategy.” In Line 344-348.

(17)L 322- it is important that it is mentioned that the studies that drive the negative effect of AG on nephrotoxicity was without plasma controls.

Response> I have created a new supplemental table explaining whether serum aminoglycoside levels were measured in each study. As you mentioned, none of the studies included in our analysis measured serum aminoglycoside levels. This represents a limitation of our study. I added the definition of renal failure in supplemental table 2.

(18)L 338 ff “prolonged infusion” – interesting points. Many/some institutions now use 24 hour pump infusions of betalactams. The comments about prolonged infusion could perhaps be moved to novel beta-lactam section as optimized or novel….

Response> From your feedaback, I added two statement in the prolonged infusion sub session. “Prolonged infusion administration strategies for intravenous beta-lactam antibiotics may include either an extended infusion (over 2 to 4 hours) or a continuous infusion (over the entire dosing interval).” In Line 400-403. “In the study on pediatric febrile neutropenia in Mexico, the targeted patients had hematological malignancies [90]. Continuous infusion showed a trend toward clinical improvement, although it did not reach statistical significance.” in Line 416-419.

 We considered moving the prolonged infusion session to the novel beta-lactams, but decided to keep it in its current location because it also applies to older agents.

(19)L 379 ref nr 36 = wrong ref should be nr 35 (please check other refs as well)!!!!

Response>Thank you for your feedback. I modified the ref number.

(20)L 447 treatment failure may be the favourable outcome.. rephrase

Response> I modified the statement of “Treatment failure may be lower  in the combination therapy of the same beta-lactam group.”

(21)L 451 we were unable to spare… rephrase

Response> I modified the statement of  “We could not reduce the use of broad-spectrum antibiotics by combining amino-glycosides with a narrow-spectrum beta-lactam”.

(22)L 513 ref 21??

Response>Thank you for your feedback. I modified the reference.

Reviewer 2 Report

Comments and Suggestions for Authors

This manuscript meta-analysis on the combination therapy of beta-lactam plus aminoglycoside in combination therapy versus beta-lactam alone in neutropenic patients. A total of 53 RCTs were selected with a clear description of the search strategy and selection process, the risk of bias was assessed with the RoB tool, and the outcome was all cause of mortality, treatment failure, and renal failure. The results suggest that there is no clear advantage in adding aminoglycoside in empirical treatment and only increasing the risk of renal failure in these patients. The English is clear, and the data are all well presented.

The mayor inconsistency that I found in your results is mainly the heterogeneity does not measure in I2 parameter. This heterogenicity most be address with stratified analysis, supporting random effect models. I suggest that this stratification analysis could be add to supplemental material or included in the results section of the manuscript. The variables for stratification are:

Time of studies.

The oldest selected study was more than 40 years ago (Doyen et al. 1983). Near of half a century, the resistance profile, the survival rate of the neutropenic patient, oncologic treatment, diagnosis, and management, some of the main variables mentioned, changed. These confounding variables are affecting in the outcomes analysis and must be addressed. I suggest that stratified the analyzes performed in periods of same risk. Another option is select only RCT with a reasonable period.

Age of patient and cancer

The clinical and survival rate is very different in hematologic malignancies in children, with an overall survival rate of 80-90% in high-income countries compared to small cell lung cancer or pancreatic cancer with a survival rate of 10-30% in five years. And all these patients are included in the same risk profile in all analysis of mortality causes. For that we strongly suggest to the authors to select a specific profile of neutropenic patients that differ in only hematologic or solid tumor and pediatric or adult patients.  

Othe considerations

-          Follow the editorial instructions for multiple figures, see example in Word template file.

-          Insert Table 2 caption.

-          Figure 3b must be changed the title in x axis.

-          Include some of the forest plot in supplemental materials, only include main results in the manuscript.

-          Figure 7 RoB results, must be previous of forest plots.

-          Consider other limitation in renal failure outcome is that the period of time of aminoglycoside is not evaluated.

Author Response

This manuscript meta-analysis on the combination therapy of beta-lactam plus aminoglycoside in combination therapy versus beta-lactam alone in neutropenic patients. A total of 53 RCTs were selected with a clear description of the search strategy and selection process, the risk of bias was assessed with the RoB tool, and the outcome was all cause of mortality, treatment failure, and renal failure. The results suggest that there is no clear advantage in adding aminoglycoside in empirical treatment and only increasing the risk of renal failure in these patients. The English is clear, and the data are all well presented.

The mayor inconsistency that I found in your results is mainly the heterogeneity does not measure in I2 parameter. This heterogenicity most be address with stratified analysis, supporting random effect models. I suggest that this stratification analysis could be add to supplemental material or included in the results section of the manuscript. The variables for stratification are:

(1)Time of studies.

The oldest selected study was more than 40 years ago (Doyen et al. 1983). Near of half a century, the resistance profile, the survival rate of the neutropenic patient, oncologic treatment, diagnosis, and management, some of the main variables mentioned, changed. These confounding variables are affecting in the outcomes analysis and must be addressed. I suggest that stratified the analyzes performed in periods of same risk. Another option is select only RCT with a reasonable period.

 (2)Age of patient and cancer

The clinical and survival rate is very different in hematologic malignancies in children, with an overall survival rate of 80-90% in high-income countries compared to small cell lung cancer or pancreatic cancer with a survival rate of 10-30% in five years. And all these patients are included in the same risk profile in all analysis of mortality causes. For that we strongly suggest to the authors to select a specific profile of neutropenic patients that differ in only hematologic or solid tumor and pediatric or adult patients.  

Response>I conducted a sensitivity analysis focusing only on randomized controlled trials with a reasonable study period. I added the sensitivity analysis in the Line159-162 “In our systematic review, we studied a mixed population, including children and adults, spanning various ages from earlier to more recent studies. Therefore, we con-ducted sensitivity analyses based on age and year of publication” and result are in the supplement figure and mentioned in Line 239-258.

Other considerations

(3)-          Follow the editorial instructions for multiple figures, see example in Word template file.

Response>Thank you for your feedback. I confirmed the example file.

(4)-          Insert Table 2 caption.

Response> I inserted the table 2 caption “Summary of finding in the beta-lactam compared to beta-lactam plus aminoglycoside for febrile neutropenia”

(5)-          Figure 3b must be changed the title in x axis.

Response>Thank you for your feedback. I modified the title in x axis.

(6)-          Include some of the forest plot in supplemental materials, only include main results in the manuscript.

Response>I have relocated the funnel plots to the supplemental figure. Additionally, I have included the sensitivity analysis in the supplemental figure.

(7)-          Figure 7 RoB results, must be previous of forest plots.

Response>Thank you for your feedback. I moved RoB results in the previous of forest plots.

(8)-          Consider other limitation in renal failure outcome is that the period of time of aminoglycoside is not evaluated.

Response> I have created a supplement table 2 explaining the duration of the aminoglycoside treatment in each study. In the case of AMINOLACTAM, the duration was two days, which resulted in a lower incidence of acute kidney injury compared to other randomized controlled trials.

I also conducted the sensitivity analysis of the Impact of Serum aminoglycoside monitoring on nephrotoxicity in any daily regimen in supplement table 6 and mentioned the result in Line 255-258. I added the definition of renal failure in supplement table 2. I added the limitation in the Line 376-383 as “In our systematic review, we observed that a significant number of the included studies failed to conduct serum aminoglycoside monitoring (Supplementary Table 2). Further-more, a sensitivity analysis was conducted on 11 RCTs monitoring serum aminoglyco-side, but the results of risk ratio for nephrotoxicity did not change. This could be be-cause the majority of studies administered the aminoglycoside  for a longer period than AMINOLACTAM study, or because there were no articles reporting the results of aminoglycoside concentration levels. Even if they were measured, it is possible that there was nephrotoxicity or efficacy for the infection, but many of the studies were old and details could not be investigated.”